# Effect of Nanoparticles of DOX and miR-125b on DNA Damage Repair in Glioma U251 Cells and Underlying Mechanisms

**DOI:** 10.3390/molecules27196201

**Published:** 2022-09-21

**Authors:** Lin Wang, Tingting Pan, Yan Wang, Jiewen Yu, Peiyi Qu, Yue Chen, Hua Xin, Sicen Wang, Junxing Liu, Yan Wu

**Affiliations:** 1School of Pharmacy, Health Science Center, Xi’an Jiaotong University, Xi’an 710061, China; 2Clinical Laboratory, The First Affiliated Hospital of Jiamusi University, Jiamusi 154003, China; 3School of Basic Medical Sciences, Jiamusi University, Jiamusi 154007, China; 4Division of Nanomedicine and Nanobiology, National Center for Nanoscience and Technology, Beijing 100190, China

**Keywords:** glioma, nanoparticles (NPs), DNA damage, miR-125b

## Abstract

Glioma is the most common primary craniocerebral malignant tumor, arising from the canceration of glial cells in the brain and spinal cord. The quality of life and prognosis of patients with this disease are still poor. Doxorubicin (DOX) is one of the most traditional and economical chemotherapeutic drugs for the treatment of glioma, but its toxic effect on normal cells and the resistance of tumor cells to DOX make the application of DOX in the treatment of glioma gradually less effective. To solve this problem, we co-encapsulated DOX and endogenous tumor suppressor miR-125b into nanoparticles (NPs) by nanoprecipitation methods, and passively targeted them into glioma cells. In vitro experiments show that miR-125b and DOX can be effectively encapsulated into nanoparticles with different ratios, and by targeting YES proto-oncogene 1 (YES1), they can affect the adenosine 5′-monophosphate (AMP)-activated protein kinase (AMPK)/p53 pathway and induce brain glioma cell apoptosis. They can also affect the DNA damage repair process and inhibit cell proliferation. The obtained data suggest that co-delivery of DOX and miR-125b could achieve synergistic effects on tumor suppression. Nanosystem-based co-delivery of tumor suppressive miRNAs and chemotherapeutic agents may be a promising combined therapeutic strategy for enhanced anti-tumor therapy.

## 1. Introduction

Glioma is the most common primary malignant intracranial tumor caused by the canceration of brain and spinal cord glial cells. It mainly originates from the glial cells constituting brain parenchyma [1]. The incidence of glioma accounts for about 60% of the total incidence of intracranial tumors [2]. Currently, the primary treatment for glioma is surgery, which is supplemented by radiotherapy, chemotherapy, and immunotherapy [3]. Surgical removal of glioma still faces the problem of an extremely high recurrence rate, possibly because of the intense stress and inflammatory reactions in surgery which induce immunosuppression and promote the local occurrence of cancer [4]. As for the poor effects of radiotherapy and chemotherapy, the most probable causes are the intrinsic or acquired resistance to radiotherapy and chemotherapy drugs and the invasion of other sites in post-operative recurrence [5]. This suggests that traditional treatment strategies should be further improved to maximize therapeutic effects.

DNA damage repair is realized via the interactions between a series of signaling pathways and enzymes. Ataxia-telangiectasia mutated (ATM) protein kinase, as a member of the kinase family, has sequence homology with phosphoinositide-3-kinase (PI3K), and constitutes the core of DNA damage repair. Its main functions include regulating the activation of checkpoints under DNA double-strand break or oxidative stress and coordinating DNA repair, cell cycle progression, and cell metabolism [6]. The pathogenic germline mutations of ATM play a role in DNA damage reactions and cell cycle checkpoints, so ATM has now become an important marker of cancers and a new target for cancer treatment [7]. Histones play a pivotal role in the composition and gene function regulation of chromatin structures in eukaryotes. The nucleosome constituting chromatin is mainly composed of four core histones (H2A, H2B, H3, and H4) and one monomer histone H1, while another protein subtype H2AX plays a critical part in DNA damage repair.

DOX is one of the most traditional and economical chemotherapy drugs for treating glioma. However, the toxic effect of DOX on normal cells and the resistance of tumor cells to DOX have reduced the use of DOX in glioma treatment. Resistance to DOX has also become a research emphasis in clinical treatment with DOX. So far, many molecular pathways and mechanisms of resistance to DOX have been identified [8]. Non-coded RNA also plays a vital role in resistance to DOX. MicroRNA-125b (miR-125b) is one of the most important miRNAs that regulate all kinds of physiological and pathological processes. Currently, the role of miR-125b in many cancers has been well confirmed. By binding with the 3′ untranslated region (3′-UTR) of target mRNA, it causes the degradation or translational inhibition of target mRNA [9]. It has been proven that miR-125b experiences expression dysregulation in many cancers [9], and affects the development of cancer by affecting cell proliferation and apoptosis-related pathways such as PI3K [10] and NF-κB [11]. Studies have shown that the expression level of miR-125b drops significantly in breast cancer patients, and that the overexpression of miR-125b enhances the sensitivity of DOX-resistant breast cancer cell MCF-7/R to DOX [12]. Therefore, we speculate that miR-125b also increases the sensitivity of tumors to DOX in other tumors, and this effect may be achieved through its downstream target genes. Through bioinformatics analysis, a possible target of miR-125b is the *yes1* gene. YES1 is a tyrosine kinase coded by proto-oncogene *yes1*. As a member of the src protein family, YES1 plays a critical part in cell proliferation, adherence, and differentiation [13].

NPs show great application potential in the medical field [14]. Compared with free drugs, NPs can directly transport drugs to specific cells or tissues, thereby greatly reducing the side effects of drugs and increasing the site-specific delivery of drugs [15]. At the same time, packaging such as nanoparticles can also reduce drug toxicity and improve efficacy. Preliminary experiments [16] have demonstrated that nanoparticles have great application potential in the delivery of antitumor drug DOX. In our present studies, nanocarriers can deliver dual drugs in addition to single drugs. Nano-complexes were prepared using nanoprecipitation which simultaneously encapsulate DOX and miR-125b mimics. The particle size, surface zeta potential, morphology, DOX and miR-125b encapsulation efficiency were characterized. The synergetic effects and mechanisms of DOX and miR-125b in glioma therapy were further investigated in vitro.

## 2. Results

### 2.1. Construction of Nanoparticles

The morphology of NPs was observed by TEM. The NPs appeared to be typical spheres in shape with good dispersion (Figure 1A). The average size, size distribution, and surface potentials of NPs, NPs+miR-125b, DOX+NPs, and DOX+NPs+miR-125b were measured via DLS (Figure 1B,C). The average size of NPs, NPs+miR-125b, DOX+ NPs, and DOX+NPs+miR-125b was 256.08 nm, 237.2664 nm, 220.7669 nm, 235.6625 nm, respectively. All the NPs showed a narrow size distribution (PDI < 0.2). The zeta potential for the NPs, NPs+miR-125b, DOX+NPs, and DOX+NPs+miR-125b was −17.8 mV, −17.5 mV, −19.9 mV and −19.1 mV, respectively (Table 1). Then we examined the encapsulation efficiency for DOX and miR-125b at different material mass ratios (Table 2). An optimal proportion of 8:1 (DOX+NPs: miR-125b, mass ratio) was chosen for the following studies.

### 2.2. miR-125b Targets the 3′UTR of YES1 mRNA

First, we measured the mRNA level of YES1 in miR-125b mimics-transfected cells (Figure 2B). As expected, the mimics of miR-125b significantly downregulated the mRNA level of YES1 when compared with control group. In order to make sure whether miR-125b can directly target YES1 3′UTR, we predicted the binding site of miR-125b to the yes1 3′UTR region (Figure 2A) and mutated this binding site (Figure 2C). Luciferase report vectors were constructed with wild-type YES1 3′UTR (YES1 3′UTR WT) and mutated YES1 3′UTR (YES1 3′UTR MT). The YES1 luciferase activity was measured for describing miR-125b function on luciferase translation. The results showed that luciferase activity of wild-type YES1 3′UTR was significantly inhibited by miR-125b overexpression, yet mutated YES1 3′UTR terminated this effect (Figure 2D). Taken together, we confirmed that YES1 is the dir000ect target of miR-125b. Thus, we reached the conclusion that miR-125b targets YES1 and regulates YES1 expression.

### 2.3. The Results of Cell Viability and Lysosome Localization

miR-125b was transfected into cells, and the cell viability of U251 cells transfected with different concentrations of miR-125b at 24 h and 48 h was determined by MTS assay (Figure 3A). It can be seen that with the increase of miR-125b concentration and duration, the viability of U251 cells gradually decreased. This result suggests that miR-125b inhibits U251 in a time- and dose-dependent manner.

U251 cells were treated with DOX, NPs, DOX+NPs, and DOX+NPs+miR-125b for 24 h and 48 h, respectively (the drug concentration is based on DOX concentration). Cell viability was determined by MTS assay (Figure 3B,C). According to the results, with the increase of DOX drug concentration and the increase of action time, the cell viability of each group decreased gradually. However, there was no difference in cell viability between NPs, DOX-NPs and the same concentration of DOX (*p* > 0.05), which indicated that nanoparticles had no effect on the viability of U251 cells. We chose the conditions (DOX concentration of 1 μg/mL, DOX+NPs and miR-125b ratio of 8:1, and the effect time of 24 h) with cell viability around 50% and statistically significant (*p* < 0.05) for our next experiments.

Cellular uptake of DOX+NPs+miR-125b was assessed by CLSM measurements. DOX+NPs+miR-125b was incubated with U251 cells at 37 °C for 0.5 and 1 h. We used the red autofluorescence of DOX and the green fluorescence of FITC to study the cellular uptake of nanoparticles and the intracellular localization of DOX. As shown in Figure 4A, a small amount of red fluorescence was observed in U251 cells incubated with DOX+NPs+miR-125b for 0.5 h, and when the cells were incubated for 1 h, strong DOX red fluorescence appeared around the cytoplasm. This suggests that DOX+NPs+miR-125b may be taken up by cells through a nonspecific endocytic mechanism, and DOX molecules are released in the endocytic compartment.

To evaluate the endosomal escape behavior of nanocarriers, DOX+NPs+miR-125b was incubated with U251 cells for different durations. At predetermined time points (0, 2, 4 h), co-localization analysis was performed to examine the extent of nanoparticle and lysosome overlap (Figure 4B). Green fluorescence indicates the location of the micellar nanocarriers. Red fluorescence from a commercial probe (LysoTracker^®^ Red DND-99, Invitrogen, Waltham, MA, USA) indicates the site of the lysosome. The overlap of green and red produces yellow due to co-localization of nanocarriers and lysosomes. The green fluorescence of NPs can be seen within the organelle by overlaying the fluorescence images of LysoTracker Red (BOC Sciences, New York, NY, USA), which is concentrated in the lysosome, with only a small fraction randomly distributed in the cytoplasm. These results suggest that delivery of DOX-loaded NPs via lysosomes is likely to enhance DOX release, which we demonstrate occurs efficiently at lysosomal pH. Regarding DOX+NPs+miR-125b, the degree of colocalization remained high over the course of the experiment (i.e., 4 h), suggesting a lower degree of endosomal escape.

### 2.4. Arresting Effect of DOX+NPs+miR-125b on U251 Cell Cycle

After treatment with different DOX concentrations of DOX, DOX+NPs and DOX+NPs+miR-125b, the cell cycle was detected by flow cytometry (Figure 5A), and all concentrations of DOX, DOX+NPs and DOX+NPs+miR-125b could induce G0/G1 arrest in U251 cells. The G0/G1 phases of U251 cells treated with DOX at concentrations of 0.25, 0.5 and 1 μg/mL were 55.13%, 61.04% and 65.18%, respectively. After treatment with DOX+NPs at DOX concentrations of 0.25, 0.5 and 1 μg/mL, the G0/G1 phase distribution of U251 cells was 54.67%, 60.43% and 64.54%, respectively. After treatment with DOX+ NPs+miR-125b at DOX concentrations of 0.25, 0.5 and 1 μg/mL, the G0/G1 phase distribution of U251 cells was 60.10%, 65.44% and 71.08%, respectively. While the untreated cells showed only 51.12% of the G0/G1 phase distribution of the treated U251 cells, the G0/G1 phase arrest was significantly higher than that of the untreated U251 cells (*p* < 0.001). It is worth noting that compared with the same concentration of DOX group, DOX+NPs+miR-125b very significantly increased the G0/G1 phase arrest of U251 cells (*p* < 0.001), while DOX+NPs and the same concentration of DOX group did not have this effect (Figure 5B). This suggests that miR-125b, but not nanoparticles, can significantly enhance the cell cycle arrest effect of DOX.

Cell cycle checkpoints were detected by Western Blot and immunofluorescence (Figure 6). Cell cycle checkpoints were detected by Western Blot in U251 cells treated with different drugs, and it was found that compared with untreated U251 cells, the expression levels of ATM, H2AX, p-ATM and γ-H2AX were increased in drug-treated cells, and the difference was statistically significant (*p* < 0.05). Compared with the DOX group, the cells treated with DOX+NPs+miR-125b also had significantly higher expression of the above proteins, and the difference was statistically significant (*p* < 0.01). However, there was no significant difference in protein expression between the DOX+NPs group and the DOX group, indicating the effect of miR-125b on aggravating DOX-induced cell cycle arrest (Figure 6A). The results of immunofluorescence assay for γ-H2AX and p-ATM were the same as those of Western Blot (Figure 6B). It can be seen from the figure that the fluorescence intensity of the control group is very weak, and the fluorescence intensity of the treated cells is higher than that of the control group. Moreover, the fluorescence intensity of cells treated with the same drug also increased with the increase of DOX concentration in the drug. After treatment with the same DOX concentration, the fluorescence intensity of DOX+NPs+miR-125b group was higher than that of DOX group, indicating that DOX+NPs+miR-125b induced more severe cell cycle arrest compared with DOX.

### 2.5. miR-125b and DOX+NPs+miR-125b Altered the Expression of YES1, AMPK, p53 and Corresponding Phosphorylated Proteins

Studies have shown that YES1 promotes the expression of YES1-associated protein (YAP1), which can also be regulated by AMPK, and its expression is increased in the presence of AMPK inhibition [16]. However, the relationship between YES1 and AMPK is still uncertain. In view of the important role of AMPK/p53 in apoptosis, we inhibited the expression of YES1 by transfecting miR-125b, detected the changes of AMPK expression, and analyzed the relationship between YES1 and AMPK. After transfecting U251 cells with different concentrations of miR-125b for 24 h (Figure 7), the expression levels of YES1 and p-YES1 gradually decreased with the increase of miR-125b concentration, and the difference was statistically significant (*p* < 0.05). Especially after the action of medium and high concentrations of miR-125b, the expression of YES1 and p-YES1 decreased more significantly (*p* < 0.01). With the increase of miR-125b concentration, the expressions of AMPK, p-AMPK, p53 and p-p53 gradually increased (*p* < 0.05), and the change of phosphorylated protein expression was more significant than that of total protein. (*p* < 0.01). From the results, we can speculate that YES1 protein may have an inhibitory effect on AMPK, and that miR-125b indirectly activates the AMPK/p53 signaling pathway and promotes apoptosis by inhibiting the expression of YES1.

U251 cells were treated with the same DOX concentration of DOX, DOX+NPs, DOX+NPs+miR125b (Figure 8), and compared with the control group. The expressions of YES1 and p-YES1 in the drug-treated group were decreased, and the difference was statistically significant (*p* < 0.05). However, AMPK, p-AMPK, p53, and p-p53 showed the opposite trend, that is, compared with the control group, the expression of the above four proteins increased in the drug treatment group (*p* < 0.05), and the changes of p-AMPK and p-p53 were more significant (*p* < 0.01). It is worth noting that compared with the DOX group, the expression levels of YES1 and p-YES1 in the DOX+NPs+miR-125b group were significantly decreased (*p* < 0.01), while the expression levels of AMPK, p-AMPK, p53 and p-p53 were significantly increased (*p* < 0.01). However, compared with the DOX group, DOX+NPs had no significant effect on protein expression (*p* > 0.05).

## 3. Discussion

Glioma is characterized by high cellular heterogeneity, fast proliferation, and strong invasiveness. Currently glioma is mainly treated by surgery, together with radiotherapy and chemotherapy, but post-operative recurrence is still unavoidable [17]. DNA damage repair is realized via the interactions between a series of signaling pathways and enzymes. Ataxia-telangiectasia mutated (ATM) protein kinase, as a member of the kinase family, has sequence homology with PI3K, and constitutes the core of DNA damage repair. Its main functions include regulating the activation of checkpoints under DNA double-strand break or oxidative stress and coordinating DNA repair, cell cycle progression, and cell metabolism [18]. The pathogenic germline mutations of ATM play a role in DNA damage reactions and cell cycle checkpoints, so ATM has now become an important marker of cancers and a new target for cancer treatment [19]. Histones play a pivotal role in the composition and gene function regulation of chromatin structures in eukaryotes. The nucleosome constituting chromatin is mainly composed of four core histones (H2A, H2B, H3, and H4) and one monomer histone H1, while another protein subtype H2AX plays a critical part in DNA damage repair.

The DOX+NPs+miR-125b NPs constructed in this study can inhibit the proliferation of glioma cells. One of the possible mechanisms lies in that NPs can induce DNA damage and inhibit DNA repair in glioma. Compared to DOX used alone, DOX+NPs+miR-125b double-loaded NPs significantly up-regulate the expression levels of γ-H2AX and p-ATM. Histone H2AX is an important cell cycle checkpoint. In case of DNA double-strand break (DSB) damage, H2AX experiences rapid phosphorylation, resulting inγ-H2AX. γ-H2AX can recruit cell cycle-related proteins and repair proteins to the damage site. These proteins and γ-H2AX form γ-H2AX foci. As one of the protein complexes recruited in the early stage after occurrence of DNA damage, γ-H2AX foci provide binding sites for other repair proteins, such as BRCA1 (breast cancer gene 1) and 53BP1 (p53 binding protein), and contribute to DSB repair. Thus, γ-H2AX is a biomarker that can characterize DNA DSB damage and repair [20]. A higher expression level of γ-H2AX suggests that DOX+NPs+miR-125b induces more serious DNA DSB damage than DOX and has a stronger lethality against tumors. Meanwhile, DNA damage initiates DNA damage response (DDR). ATM, as the core protein of DNA damage repair, automatically experiences phosphorylation and further phosphorylates downstream target cell cycle checkpoint kinase 2 (CHK2). CHK2 is a protein kinase, and one important substrate of CHK2 is cell division cycle 25 homolog A (CDC25A). CDC25A activates cyclin-dependent kinase 2 (CDK2), and promotes cells to develop from G1 phase to S phase. However, when CDC25A is phosphorylated by p-CHK2, its activity declines, and its functions are inhibited, making it impossible for cells to enter S phase and resulting in G0/G1 phase arrest. As a result, DNA damage cannot be normally repaired, and the proliferation of tumor cells is inhibited.

YES1 is a tyrosine kinase coded by proto-oncogene *yes1*. As a member of the src protein family, YES1 plays a critical part in cell proliferation, adherence, and differentiation [21]. Dual-luciferase detection experiments have confirmed the role of yes1 as a target gene of miR-125b, and shown that miR-125b negatively regulates the expression of YES1. Adenosine monophosphate-activated protein kinase (AMPK) is a critical molecule in bioenergy metabolism regulation, and a centrin linking up anabolism and catabolism. Its role in diabetes and other metabolic diseases is widely known. However, recent studies have revealed that AMPK also plays a significant role in the regulation of apoptosis, autophagy, and cell cycle [22]. Activation of AMPK/p53 pathway is important for apoptosis.

In this study, in control U251 cells, the expression levels of YES1 and p-YES1 were relatively high, while the expression levels of p-AMPK and p-p53 were relatively low. After miR-125b transfected cells, the expression of YES1 decreased under the action of miR-125b because of its negative regulation of the expression of YES1, and the expression level gradually decreased with the increase of miR-125b concentration. At the same time, the expression levels of p-AMPK and p-p53 increased with the increase of miR-125b concentration (Figure 7). According to the experimental results, we have reason to believe that YES1 has an inhibitory effect on AMPK, and miR-125b alleviates its inhibitory effect on AMPK by inhibiting the expression of YES1, resulting in an increase in the active form of AMPK (p-AMPK), thereby activating AMPK/p53 pathway and induction of apoptosis. We used our constructed double-loaded nanoparticles to act on U251 cells, and the results showed (Figure 8) that although DOX can also activate the AMPK/p53 pathway by inhibiting the expression of YES1, the effect is far less than that of DOX and miR-125b. Compared with DOX or mi-125b alone, dual-loaded nanoparticles inhibited YES1 and activated the AMPK/p53 pathway more significantly, and induced tumor cell apoptosis more strongly.

In conclusion, miR-125b can aggravate the cell cycle arrest caused by DNA damage by DOX, and may also attenuate its inhibitory effect on AMPK by inhibiting YES1, thereby activating the AMPK/p53 pathway. AMPK/p53 pathway plays an important role in apoptosis. Therefore, we speculate that miR-125b can promote the apoptosis of U251 cells by inhibiting YES1 to activate the AMPK/p53 pathway, but further experiments are needed to prove this. Compared with DOX or miR-125b alone, the double-loaded nanoparticles combined with DOX and miR-125b could not only accurately enter tumor cells, but also had stronger activation effects on DNA damage and apoptosis pathways.

## 4. Materials and Methods

### 4.1. Drugs and Reagents

DOX was provided by Beijing Huafeng United Technology (Beijing, China). Antibodies against GAPDH, ATM, p-ATM, H2AX, p-H2AX, YES1, p-YES1, p53, p-p53, AMPKα, p-AMPKα were purchased from Abcam (Cambridge, UK). Dulbecco’s modified Eagle’s high glucose medium and Dulbecco’s modified Eagle’s medium were provided by GIBCO (Grand Island, New York, NY, USA). Dual-Luciferase Reporter Assay System Kit (Thermo Fisher Scientific, Waltham, MA, USA) and pSI-CHECK2 vector were provided by Hanbio Biotechnology (Wuhan, China). EntiLink™ 1st Strand cDNA Synthesis Kit and EnTurbo™ SYBR Green PCR SuperMix Kit were purchased from ELK Biotechnology (Wuhan, China).

### 4.2. Construction of NPs

DOX and miR-125b were encapsulated into copolymer NPs by the nano-precipitation method, as detailed in previous research [16]. The sizes, distribution, and surface potentials of prepared NPs were measured by dynamic light scattering (DLS).

### 4.3. Cell Lines and Cell Culture

Human glioma cells U251 and human embryonic kidney cell line 293T were from Shanghai Cell Bank (Shanghai, China), Chinese Academy of Science. U251 was cultured in Dulbecco’s modified Eagle’s high glucose medium with 10% fetal bovine serum (FBS) and 1% penicillin/streptomycin. To investigate the role of DOX-miR-125b-NPs in glioma, human glioma U251 cells were exposed to a certain concentration of DOX, DOX NPs, miR-125b, DOX-miR-25b-NPs for a period of time. The 293T was cultured in DMEM used for the dual-luciferase assays. The medium was supplemented with 10% fetal bovine serum (FBS) and 1% penicillin/streptomycin. Cells were cultivated in a humidified incubator containing 5% CO_2_ at 37 °C.

### 4.4. Cell Viability Assay

The proliferation effects of drugs on U251 cells were determined by 3-(4,5-dimethylthiazol-2-yl)-5-(3-carboxymethoxyphenyl)-2-(4-soufophenyl)-2H-tetrazolium, inner salt(MTS) assay. In brief, cells were plated in a 96-well plate overnight to adhere, then different drugs were administrated and incubation continued for 24 or 48 h. MTS solution (20 µL per well) was added and incubated for another 4 h at 37 °C, discarding the super natant and using dimethyl sulfoxide (DMSO) to dissolve products for 10 min at 37 °C. Microplate reader (BioTek, Winooski, VT, USA) was used to measure the 96-well plate at 490 nm, and the cell viability (%) was calculated by OD values.

### 4.5. Dual-Luciferase Detection Experiment

For the luciferase reporter assay, the 3′UTR of YES1 containing the wild or mutant miR-125b target sites was cloned using primers with NotI and XhoI cleavage sites. The wild or mutant type 3′UTR fragment was inserted into the corresponding site of the pSI-CHECK2 vector and then co-transfected into 293T cells with miR-125b mimics/mock. After 48 h transfection, the cells were harvested and the Dual-Luciferase Reporter Assay System Kit was used for detecting dual-luciferase activity, according to the manufacturer’s instructions.

### 4.6. Real Time-Polymerase Chain Reaction

Total RNA was isolated using the TRIzol reagent (ELK Biotechnology, Wuhan, China) according to the manufacturer’s protocol. cDNA was synthesized using the EntiLink™ 1st Strand cDNA Synthesis Kit. All real-time polymerase chain reactions (PCRs) were performed using EnTurbo™ SYBR Green PCR SuperMix Kit on a StepOne™ Real-Time PCR system (Life Technologies, New York, NY, USA). GAPDH was used as an internal control for normalization of the relative expression levels. Gene expression levels were calculated using the 2^–ΔΔCT^ method relative to that of the internal control. Primers are listed in Table 3.

### 4.7. Western Blot

U251 were treated with different concentrations of drugs. Cells were centrifuged, then washed twice with PBS and lysed with loading buffer for 45 min in 4 °C. Then, the cell lysate was boiled for 10 min and stored at −80 °C. Protein samples were resolved by 8–10% SDS-PAGE, transferred to PVDF membranes (Millipore, Billerica, MA, USA). The PVDF membranes were blocked with 5% skimmed milk and then incubated with the primary antibodies overnight at 4 °C. Subsequently, the membranes were incubated with HRP-secondary antibody at 37 °C for 1 h. Finally, the image was detected by Tanon 5200 (Tanon, Beijing, China).

### 4.8. Flow Cytometry

Cells were harvested and fixed for 20 min at 4 °C in 90% ethanol. Thereafter, the cells were washed twice with phosphate buffered saline (PBS) and then stained with the PI/RNase staining buffer (Sungene Biotech, Shanghai, China). Cell-cycle distribution was determined using flow cytometry. Each experiment was repeated three times.

### 4.9. Laser Confocal Microscopy

#### 4.9.1. Drug Uptake and Lysosomal Colocalization in U251 Cells

Cells were inoculated in a six-well culture plate at the density of 1 × 10^5^ cells/well for 24 h, and each well was placed with a sterile coverslip. After cell adherence, culture medium was added with fluorescein isothiocyanate (FITC)-labeled free drugs for culture at 37 °C for certain time, respectively. For the purpose of detecting intracellular localization, cells were further incubated with lysosomal red fluorescent probe 75 nM LysoTracker Red DND-99 for 30 min for lysosome labeling. The supernatant was carefully removed, and cells were washed three times with precooled PBS. Samples were detected by CLSM using Olympus FV1000 (Olympus, Shibuya, Japan). kex 488 nm and kem 510 nm were used for LysoTracker (BOC Sciences, New York, NY, USA) and FITC-labeled NPs (lysosomal red fluorescent probe and FITC; λex 488 nm and λem 510 nm).

#### 4.9.2. Immunofluorescence

Cells were inoculated in a six-well culture plate at the density of 1 × 10^5^ cells/well for 24 h, and each well was placed with a sterile coverslip. After cell adherence, cells were washed once with PBS and fixed with 4% formaldehyde for 15 min, followed by three times of washing with PBS (5 min each time). After 10 min of permeation at room temperature (0.2%), cells were washed three times with PBS (5 min each time), blocked with 5% BSA+PBS at room temperature for 45 min, and incubated in 4 °C primary antibody overnight. After that, they were washed three times with PBS (5 min each time), and incubated in fluorescent secondary antibody for 30–40 min in dark place, followed by three times of washing with PBS (15 min each time). After addition of 30–50 µL DAPI dye liquor to each well, staining continued for 3–5 min. Finally, cells were washed three times with PBS (5 min each time), sealed, and observed under a laser confocal scanning microscope.

### 4.10. Statistical Analysis

All data were presented as mean ± standard deviation. Experiments in each group were repeated at least three times. The statistical significance between two groups was analyzed using *t*-test. Intergroup significance was determined by one-way analysis of variance (ANOVA). The difference was considered statistically significant when * *p* < 0.05, ** *p* < 0.01, *** *p* < 0.001.

## 5. Conclusions

In this study, we report the successful application of DOX+NPs+miR-125b prepared by nanoprecipitation method. Our data showed that DOX+NPs+miR-125b inhibited U251 more significantly than DOX alone. The reason is that miR-125b can aggravate the DNA damage caused by DOX, leading to more severe cell cycle arrest; it can also attenuate the inhibitory effect of YES1 on the AMPK/p53 pathway and promote cell apoptosis. Drugs and miRNAs are linked together by nanoparticles, which can not only amplify the effect of the drug itself, but also bring the corresponding role of miRNAs into play. Since the co-delivery method has the advantage of simultaneously inhibiting tumor growth and migration, the co-delivery of miRNAs and chemotherapeutic drugs through nanosystems has shown great potential as a combination therapy strategy in anticancer therapy. Owing to the advantages of the co-delivery approach for the simultaneous inhibition of tumor growth, co-delivery of miRNAs and chemotherapeutic drugs by nano-systems demonstrates a great potential as combined therapeutic strategy in anti-cancer treatment. It provides new ideas for research and development into tumor drugs, especially to improve the sensitivity of anti-tumor drugs.

## Figures and Tables

**Figure 1 molecules-27-06201-f001:**
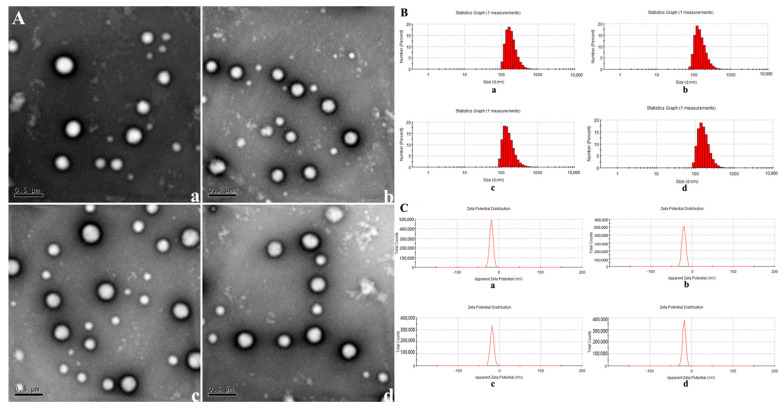
(**A**) TEM images of nanoparticles. (**a**): NPs, (**b**): NPs+DOX, (**c**): NPs+miR-125b, (**d**): DOX+NPs+miR-125b. (**B**) DLS measurement of nanoparticle diameter. (**a**): NPs, (**b**): NPs+DOX, (**c**): NPs+miR-125b, (**d**): DOX+NPs+miR-125b. (**C**) DLS measurement of nanoparticle charges. (**a**): NPs, (**b**): NPs+DOX, (**c**): NPs+miR-125b, (**d**): DOX+NPs+miR-125b.

**Figure 2 molecules-27-06201-f002:**
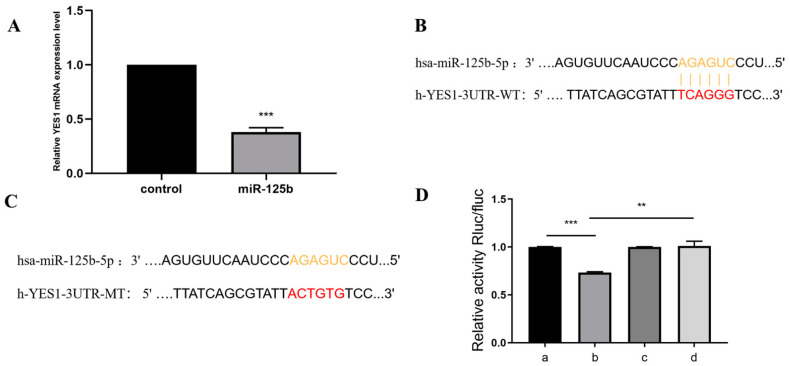
miR-125b directly binds to YES1 3′UTR. (**A**) qRT-PCR analysis of levels of gene expression in YES1 with miR-125b mimics or control. (**B**) The predicted miR-125b binding site in YES1 3′UTR. (**C**) mutant miR-125b binding site in YES1 3′UTR. (**D**) Luciferase assay of transfected with wild-type or mutant YES1 3′UTR plasmid in 293T cells. a: NC mimics+h-yes1-3UTR-wt; b: hsa-miR-125b-5p+ h-yes1-3UTR-wt; c: NC mimics+h-yes1-3UTR-mu; d: hsa-miR-125b-5p+h-yes1-3UTR-mu. *N* ≥ 3 for B and D, ** indicates *p* values < 0.01, *** indicates *p* values < 0.001.

**Figure 3 molecules-27-06201-f003:**
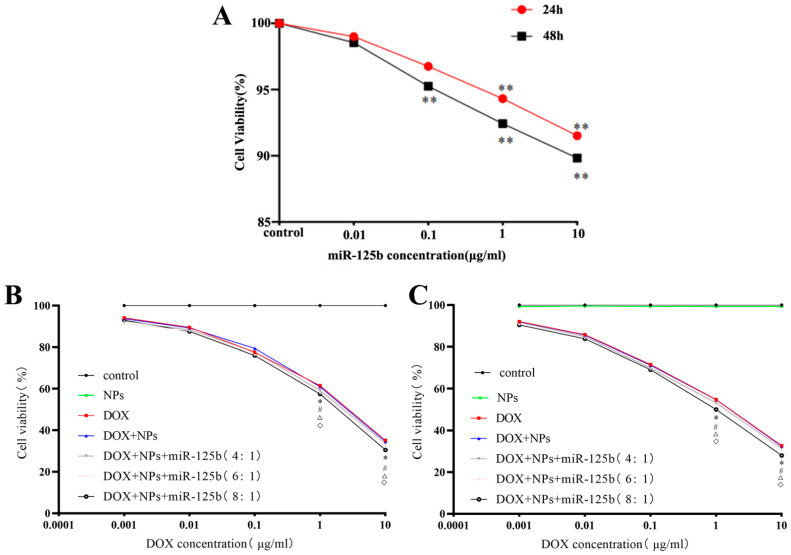
(**A**) The survival rates of U251 cells under different miR-125b concentrations. (**B**) 24 h survival rates of U251 cells under the actions of DOX, DOX+NPs, and DOX+NPs+miR-125b with different concentrations. (**C**) 48 h survival rates of U251 cells under the actions of DOX, DOX+NPs, and DOX+NPs+miR-125b with different concentrations. *N* ≥ 3 for A, B and C. * compared to the control group, *p* < 0.05, ** compared to the control group, *p* < 0.01, # compared to the NPs group, *p* < 0.05, △ compared to the DOX group with the same DOX concentration, *p* < 0.05, ◇ compared to the DOX+NPs group with the same DOX concentration, *p* < 0.05.

**Figure 4 molecules-27-06201-f004:**
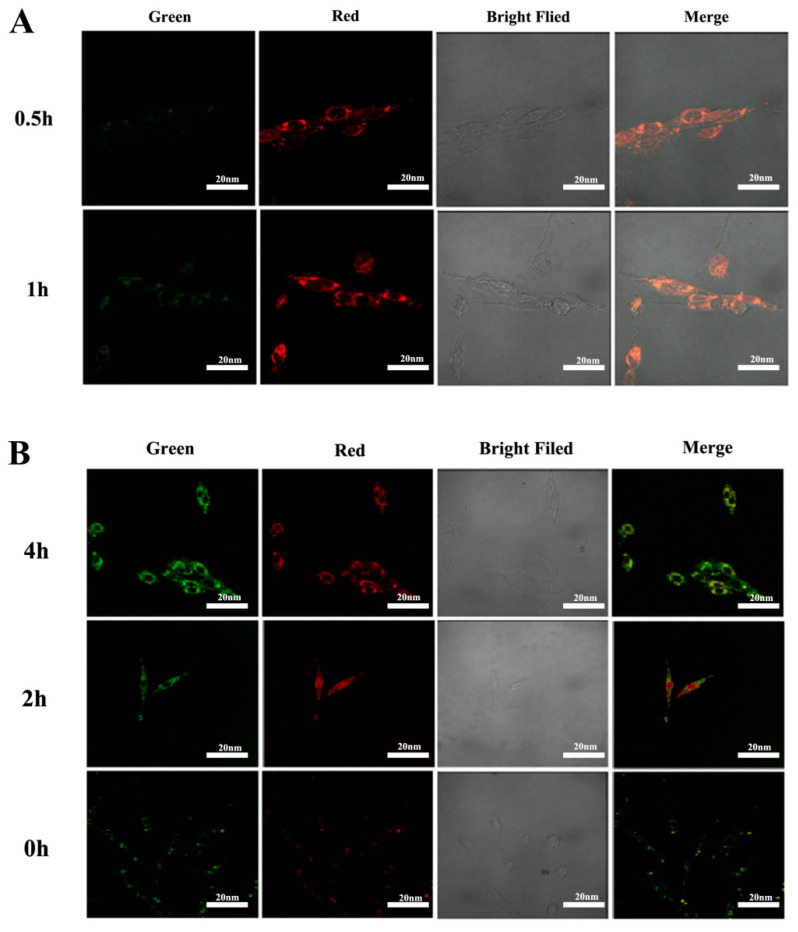
(**A**) Expression results of miR-125b in cells. (**B**) Co-localization results of NPs and lysosomes in cells.

**Figure 5 molecules-27-06201-f005:**
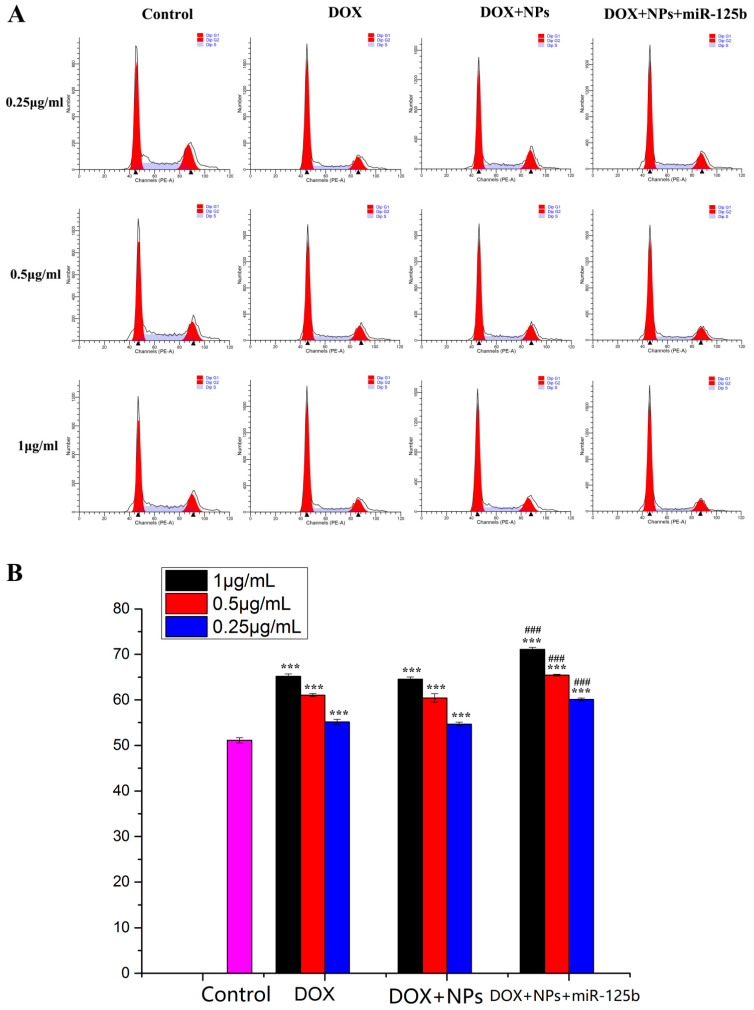
(**A**) The results of flow cytometry analysis of cell cycle. (**B**) The results of cell cycle arrest in G0/G1 phase. *N* ≥ 3 for B, *** compared to the control group with the same DOX concentration, *p* < 0.001, ### compared to the DOX group with the same DOX concentration, *p* < 0.001.

**Figure 6 molecules-27-06201-f006:**
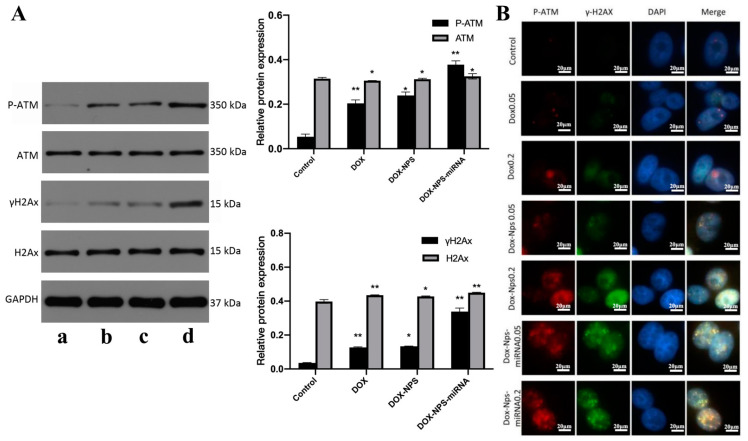
(**A**) The results of detection of expression levels of DNA damage proteins and their active forms (ATM, p-ATM, H2AX, γ-H2AX), a: control; b: DOX; c: DOX+NPs; d: DOX+NPs+miR-125b. (**B**) The results of immunofluorescence detection of p-ATM and γ-H2AX (600×). *N* ≥ 3 for A, * compared to the control group, *p* < 0.05, ** compared to the control group, *p* < 0.01.

**Figure 7 molecules-27-06201-f007:**
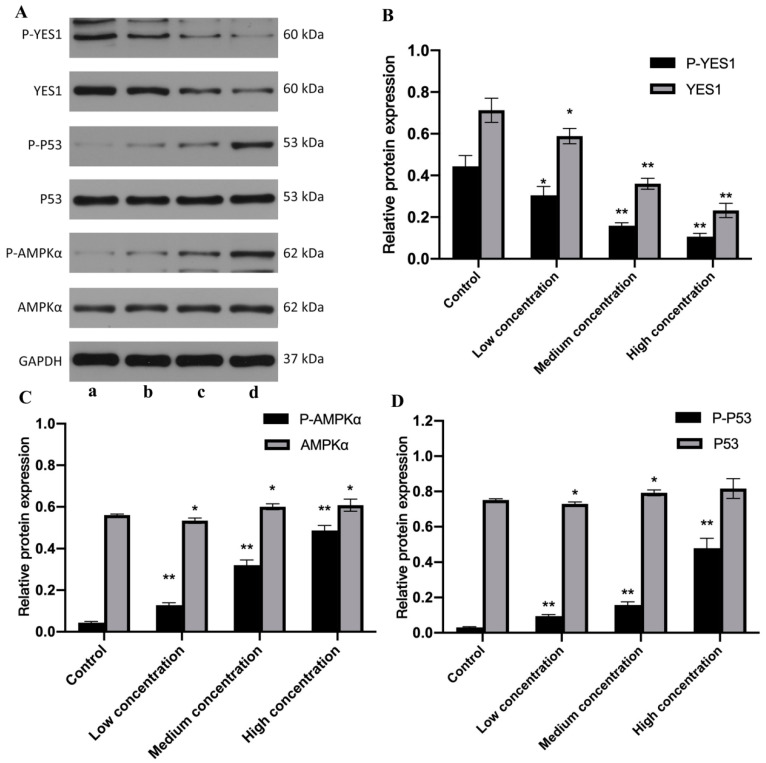
The detection results of the effect of miR-125b on the expression levels of YES1, AMPK, p53 and corresponding phosphorylated proteins. (**A**) The results of Western Blot experimental. a: control; b: low concentration; c: medium concentration; d: high concentration. (**B**–**D**) The results of Band Analysis. *N* ≥ 3 for B, C and D, * compared to the control group, *p* < 0.05, ** compared to the control group, *p* < 0.01.

**Figure 8 molecules-27-06201-f008:**
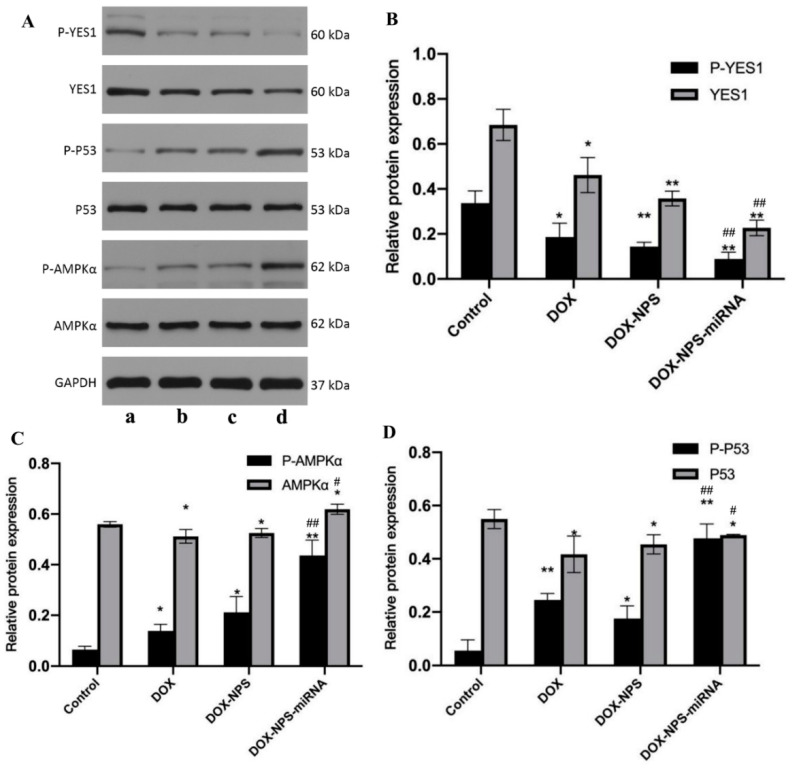
The detection results of the effect of DOX+NPs+miR-125b on the expression levels of YES1, AMPK, p53 and corresponding phosphorylated proteins. (**A**) The results of Western Blot experimental. a: control; b: DOX; c: DOX+NPs; d: DOX+NPs+miR-125b. (**B**–**D**) The results of Band Analysis. *N* ≥ 3 for B, C and D, * compared to the control group, *p* < 0.05, ** compared to the control group, *p* < 0.01, # compared to the control group, *p* < 0.05, ## compared to the DOX group, *p* < 0.01.

**Table 1 molecules-27-06201-t001:** Physicochemical characteristics of the various NPs ^a^.

Physicochemical Characteristics	Size (nm)	PDI	Zeta Potential (mV)
NPs	256.08 ± 1.68	0.069 ± 0.03	−17.8 ± 4.26
NPs+DOX	237.2664 ± 3.18	0.111 ± 0.02	−17.5 ± 4.36
NPs+miR-125b	220.7669 ± 0.71	0.056 ± 0.04	−19.9 ± 4.47
DOX+NPs+miR-125b	235.6625 ± 1.16	0.062 ± 0.03	−19.1 ± 4.7

^a^ Determined by dynamic light scattering (DLS). The NPs were prepared by directly dissolving in distilled water at a concentration of 0.5 mg/mL, followed by 5 min sonication. Results are means ± SD (*N* ≥ 3).

**Table 2 molecules-27-06201-t002:** The influences of formulation parameters on drug loading content and encapsulation efficiency.

DOX+NPs/miR-125b Mass Ratio	4/1	6/1	8/1
Encapsulation efficiency of DOX (%)	35.2 ± 2.9	40.9 ± 2.5	48.8 ± 3.2
Encapsulation efficiency of miR-125b (%)	85	89	92

**Table 3 molecules-27-06201-t003:** Primer sequence.

Primer Name	Primer InforMation	Base Sequence (5′–3′)		Tm Value	CG%	Product Length
H-GAPDH	NM_001256799.2	Sense	CATCATCCCTGCCTCTACTGG	59.4	57.1	259
		Antisense	GTGGGTGTCGCTGTTGAAGTC	60.1	57.1	
H-YES1	NM_005433.4	Sense	TTTGTGGCCTTATATGATTATGAAG	58.2	32	193
		Antisense	AATACCATTCTTCTGCCTGAATG	58.2	39.1	

## Data Availability

Not applicable.

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
