# Peer review of "Effect of Nanoparticles of DOX and miR-125b on DNA Damage Repair in Glioma U251 Cells and Underlying Mechanisms"

_molecules, 2022, doi:10.3390/molecules27196201_

Round 1
Reviewer 1 Report
The paper provides important informations about Effect of Nanoparticles of DOX and miR-125b on DNA damage repair in glioma U251 cells and underlying mechanisms. Compared with DOX or miR-125b alone, the double-loaded nano-315 particles combined with DOX and miR-125b could not only accurately enter tumor cells, 316 but also had stronger activation effects on DNA damage and apoptosis pathways.The structure of the article is well collated, following a logical and easy to interpret. In addition, tables and figures are clear, easy interpret and allows the reader to quickly locate what they are trying to determine. However, there are a few comment I would like to make for further clarification and scrutiny.
All abbreviations in the manuscript should be checked, and all terms should be used in long form when their first use. For example, miR-125b, DOX, YES1 etc.
Line 83;The aim of the study must be given more clearly at the end of the introduction section.
Line 84; Most information must be given Results of the construction of Nanoparticles. Now this section is too poor.
Line 100; Please give the in information in order to picture numbers. Firstly Fig 2A must be explained then the others.
Line 120; It can be performed statistical analysis of time and dose dependent manner by correlation analysis.
Others are shown the attached manuscript.

Author Response
Dear Reviewer:
The article has been revised and the following is in response to your comments.
Line 83;The aim of the study must be given more clearly at the end of the introduction section.
Anwser: Relevant changes have been made to the Introduction. Thanks for your suggestion.
Line 84; Most information must be given Results of the construction of Nanoparticles. Now this section is too poor.
Anwser: Nanoparticle TEM size distribution images have been added to Figure 1, and DLS and zeta potential values as well as DOX and miR-125b coverage are shown in Tables 1 and 2. The revised Figure 1, Tables 1 and 2 can be viewed in the attachment.Thanks for your suggestion.
Line 100; Please give the in information in order to picture numbers. Firstly Fig 2A must be explained then the others.
Anwser: The order of Figure 2 has been modified, and the modified Figure 2 can be viewed in the attachment.Thanks for your suggestion.
Line 120; It can be performed statistical analysis of time and dose dependent manner by correlation analysis.
Anwser: Related content has been modified. Thanks for your suggestion.
Others are shown the attached manuscript.
Anwser: The marked section of the manuscript has been revised accordingly.Thanks for your suggestion.
Thank you very much for your valuable comments on this article, the revised manuscript and pictures have been submitted.
Thank you for reviewing it again. If you need to modify it, please feel free to submit it. Looking forward to your reply again.
Kind regards,
Best regards,
Tingting Pan
Reviewer 2 Report
The authors presented the paper "Effect of Nanoparticles of DOX and miR-125b on DNA damage repair in glioma U251 cells and underlying mechanisms"
1) I think it should be mentioned in the title, abstract, results, and discussion section about what kind of the nanoparticles the text goes.
2) Moreover, some precise information about this type of the nanoparticles and possible DOX loading should be mentioned in the Introduction. As I have seen that it is Polymer-Lipid nanoparticles (ref 13), there is many such kinds DOX-loaded systems. It should be added the references to this possible systems to compare your results. Moreover, some links to the dual-loaded systems should be presented, too.
3) I recommend changing the end of the Introduction and to mention in the precise way what have been done in the work. Moreover, the Conclusion is too short and do not present any novelty of the work and limitations.
4) The DLS pictures is very good. However, it will be better to summarize the DLS and zeta potential measurements in Table. Moreover, some discussion should be presented what the size and zeta potential you need for your work.
5) Figure 3. NPs control has to be presented. Why you have no difference between DOX+NPs and DOX+NPs+miR-125b? The system is not working?
Minor comments
Abstract: line 19 yes1 should be capitalized and decrypted. Abbreviation MTS, DOX, AMPK should be decrypted, too. MTS = MTT test?
Figure 3 Where is the errors?
Figure 6A should be enlarged.
Figure 5 x-axis should be larger. It is difficult to see anything there.
Author Response
Dear Reviewer:
The article has been revised and the following is in response to your comments.
- I think it should be mentioned in the title, abstract, results, and discussion section about what kind of the nanoparticles the text goes.
Anwser: Added nanoparticle types in sections. Thanks for your suggestion.
- Moreover, some precise information about this type of the nanoparticles and possible DOX loading should be mentioned in the Introduction. As I have seen that it is Polymer-Lipid nanoparticles (ref 13), there is many such kinds DOX-loaded systems. It should be added the references to this possible systems to compare your results. Moreover, some links to the dual-loaded systems should be presented, too.
Anwser: Relevant nanoparticle information has been added to the last paragraph of the Introduction. Thanks for your suggestion.
- I recommend changing the end of the Introduction and to mention in the precise way what have been done in the work. Moreover, the Conclusion is too short and do not present any novelty of the work and limitations.
Anwser: Relevant changes have been made to the Introduction and Conclusions. Thanks for your suggestion.
- The DLS pictures is very good. However, it will be better to summarize the DLS and zeta potential measurements in Table. Moreover, some discussion should be presented what the size and zeta potential you need for your work.
Anwser: DLS and zeta potential measurements have been added to Table 1 and can be viewed in Annex 1.Thanks for your suggestion.
- Figure 3. NPs control has to be presented. Why you have no difference between DOX+NPs and DOX+NPs+miR-125b? The system is not working?
Anwser: The NPs control group has been added to Figure 3. The DOX+NPs+miR-125b group is different from the DOX+NPs group, and the corresponding statistical markers have also been added to Figure 3. the modified Figure 3 can be viewed in the attachment. Thanks for your suggestion.
Minor comments
Abstract: line 19 yes1 should be capitalized and decrypted. Abbreviation MTS, DOX, AMPK should be decrypted, too. MTS = MTT test?
Anwser: The corresponding abbreviations are used in their full names at the first mention. The MTS assay is an assay that measures cell viability like the MTT assay.
Thanks for your suggestion.
Figure 3 Where is the errors?
Anwser: The modified Figure 3 can be viewed in the attachment. Thanks for your suggestion.
Figure 6A should be enlarged.
Anwser: The figure has been enlarged, the modified Figure 6 can be viewed in the attachment. Thanks for your suggestion.
Figure 5 x-axis should be larger. It is difficult to see anything there.
Anwser:The figure has been enlarged, the modified Figure 5 can be viewed in the attachment. Thanks for your suggestion.
Thank you very much for your valuable comments on this article, the revised manuscript and pictures have been submitted.
Thank you for reviewing it again. If you need to modify it, please feel free to submit it. Looking forward to your reply again.
Kind regards,
Best regards,
Tingting Pan

Reviewer 3 Report
It is a well-designed and written paper and carries considerable merits, but some points need an explanation before the manuscript is ready for publication:
1. The introduction and discussion should focus more on this study's observations and novelty. More concluding remarks must be also added.
2. TEM with particles size distribution is required
3. Other populations for phases S and G2 must be added to figure 5 A
4. Scale bars for Figure 6 B
Author Response
Dear Reviewer:
The article has been revised and the following is in response to your comments.
- The introduction and discussion should focus more on this study's observations and novelty. More concluding remarks must be also added.
Anwser: Relevant changes have been made to the Introduction and Conclusions. Thanks for your suggestion.
- TEM with particles size distribution is required
Anwser: TEM with particles size distribution was Added to Figure 1. The modified Figure 3 can be viewed in the attachment. Thanks for your suggestion.
- Other populations for phases S and G2 must be added to figure 5 A
Anwser: The modified Figure 5 can be viewed in the attachment. Thanks for your suggestion.
- Scale bars for Figure 6 B
Anwser: A scale bar has been added to Figure 6. The modified Figure 6 can be viewed in the attachment. Thanks for your suggestion.
Thank you very much for your valuable comments on this article, the revised manuscript and pictures have been submitted.
Thank you for reviewing it again. If you need to modify it, please feel free to submit it. Looking forward to your reply again.
Kind regards,
Best regards,
Tingting Pan
Round 2
Reviewer 2 Report
Thank you for the revised paper.
However, I recommend enlarging the reference list for introduction. Maybe, to cite some review paper from MDPI system concerning DOX-loaded nanoparticles and/or glioma cancer.
For example, https://www.mdpi.com/search?sort=reference_citedby_number_max&page_count=50&year_from=2022&year_to=2022&subjects=med-pharma%2Cbio-life%2Cphysics-astronomy&journals=pharmaceutics%2Cnanomaterials%2Cmolecules%2Cpharmaceuticals%2Cbiomedicines%2Cmagnetochemistry&article_types=research-article&q=doxorubicin+nanoparticle&view=default
Author Response
Dear Reviewer:
Thank you for your affirmation of this article. Related citations have been added, please check.
Sincerely,
Tingting Pan

Reviewer 3 Report
Accept in current form
Author Response
Dear Reviewer:
Thank you for your affirmation of this article.
Best regards,
Tingting Pan
